 elife.elifesciences.org

# Structure of a pore-blocking toxin in complex with a eukaryotic voltage-dependent K+ channel

**Anirban Banerjee[1,2][†], Alice Lee[1,2], Ernest Campbell[1,2], Roderick MacKinnon[1,2]\***

[1]Laboratory of Molecular Neurobiology and Biophysics, Rockefeller University, New York, United States; [2]Howard Hughes Medical Institute, Rockefeller University, New York, United States

**Abstract** Pore-blocking toxins inhibit voltage-dependent K+ channels ($K_v$ channels) by plugging the ion-conduction pathway. We have solved the crystal structure of paddle chimera, a $K_v$ channel in complex with charybdotoxin (CTX), a pore-blocking toxin. The toxin binds to the extracellular pore entryway without producing discernable alteration of the selectivity filter structure and is oriented to project its Lys27 into the pore. The most extracellular K+ binding site (S1) is devoid of K+ electron-density when wild-type CTX is bound, but K+ density is present to some extent in a Lys27Met mutant. In crystals with Cs+ replacing K+, S1 electron-density is present even in the presence of Lys27, a finding compatible with the differential effects of Cs+ vs K+ on CTX affinity for the channel. Together, these results show that CTX binds to a K+ channel in a lock and key manner and interacts directly with conducting ions inside the selectivity filter.

**\*For correspondence:**
mackinn@rockefeller.edu

[†]**Present address:** Cell Biology and Metabolism Program, Eunice Kennedy Shriver National Institute of Child Health and Human Development, National Institutes of Health, Bethesda, United States

**Competing interests:** The authors declare that no competing interests exist.

**Reviewing editor**: John Kuriyan, University of California, Berkeley, United States

## Introduction

Poisonous animals such as tarantula spiders, green mamba snakes, or the deathstalker scorpion rely on their venom for efficient defense and precapture strategies. These venoms are stored in dedicated glands and are rapidly delivered through specialized apparatus via subcutaneous, intramuscular, or intravenous routes. What is the underlying cause of toxicity of these venoms? The principal toxic components in all of them are peptidic in nature. These venoms usually contain libraries of hundreds of peptide-based toxins that together encompass a high degree of stereochemical diversity (*Han et al., 2008*; *Liang, 2008*; *Rodriguez de la Vega et al., 2010*). Only a small fraction of these molecules, however, have been pharmacologically characterized thus far. The targets of these toxins are typically a variety of ion channels—voltage-gated Na+($Na_v$), K+($K_v$), and Ca2+($Ca_v$) channels, and cell-surface 'receptor' ion channels, such as the nicotinic acetylcholine (Ach) receptor (*Billen et al., 2008*; *King et al., 2008*; *Mouhat et al., 2008*; *Kasheverov et al., 2009*). The remarkable molecular diversity of these toxins is borne out by the fact that multiple different toxins can target different components of the same ion channel/receptor. However, the end result is alteration of the normal physiology of the ion channel/receptor, thereby eliciting the desired reaction of the venom.

Potassium channels (K+ channels), a large and diverse class of ion channels, are targets of a large number of toxins that have been characterized up to now (*Carbone et al., 1982*; *Miller et al., 1985*; *Galvez et al., 1990*; *Garcia et al., 1994*; *Swartz and MacKinnon, 1995*). Most K+ channels are tetrameric in architecture—four pore domains together form an ion-conduction pathway through the membrane (*Figure 1A*; *MacKinnon, 1991*; *Doyle et al., 1998*). In addition, in the voltage-gated family of K+ channels ($K_v$ channels), each channel monomer has attached onto the N-terminal end of the pore domain a transmembrane voltage sensor domain that senses the transmembrane voltage difference (*Figure 1A*; *Papazian et al., 1987*; *Jiang et al., 2003*). $K_v$ channels are targeted by toxins primarily at two distinct

**eLife digest** The deadly toxins produced by many creatures, including spiders, snakes, and scorpions, work by blocking the ion channels that are essential for the normal operation of many different types of cells. Ion channels are proteins and, as their name suggests, they allow ions—usually sodium, potassium, or calcium ions—to move in and out of cells. They are especially important for cells that generate or respond to electrical signals, such as neurons and the cells in heart muscle.

Ion channels are located in the lipid membranes that surround all cells, and the ions enter or leave the cell via a pore that runs through the channel protein. They can be opened and closed (or 'gated') in different ways: some ion channels open and close in response to voltages, whereas others are gated by biomolecules, such as neurotransmitters, that bind to them.

Now, Banerjee et al. have used x-ray crystallography to study the structure of the complex that is formed when charybdotoxin (CTX), a toxin that is found in scorpion venom, blocks a voltage-gated potassium channel. Previous studies have shown that CTX binds to the channel on the extracellular side of the pore. Banerjee et al. show that the toxin fits into the entrance to the channel like a key into a lock, which means the toxin is preformed to fit the shape of the channel.

The potassium ion channel is made up of four subunits, and the pore contains four ion-binding sites that form a 'selectivity filter': it is this filter that ensures that only potassium ions can pass through the channel when it is open. When CTX binds to the channel, a lysine residue poised at a critical position on the toxin is so close to the outermost ion-binding site that it prevents potassium ions binding to the site. The structure determined by Banerjee et al. explains many previous findings, including the fact that ions entering the pore from inside the cell can disrupt the binding between the toxin and the ion channel protein. It remains to be seen if the toxins that target the pore of other types of ion channels work in the same way.

sites—pore-blocking toxins that bind at the extracellular mouth of pore domains and gating-modifier toxins that bind to voltage sensor domains (*Figure 1A*; *MacKinnon et al., 1990*; *Goldstein et al., 1994*; *Swartz and MacKinnon, 1997*).

Scorpion venom specifically has been an abundant source of pore-blocking toxins for K⁺ channels. These are small peptides, typically ranging from 30 to 40 residues in length, held together by three or four disulfide bonds in a rigid architecture (*Figure 1B*; *Bontems et al., 1991*; *Johnson and Sugg, 1992*; *Fernandez et al., 1994*). Pore-blocking toxins have profoundly impacted research in the K⁺ channel field primarily in two ways. First, they have enabled purification of specific novel K⁺ channels such as the BK channel, a $Ca^{2+}$ and voltage-gated K⁺ channel (*Garcia et al., 1997*). Second, they provided our first knowledge about channel subunit stoichiometry and the shape of the extracellular K⁺ pore entryway at a time when no three-dimensional structure was available for any ion channel (*MacKinnon, 1991*; *Goldstein et al., 1994*; *Gross et al., 1994*; *Stampe et al., 1994*; *Hidalgo and MacKinnon, 1995*; *Naranjo and Miller, 1996*; *Ranganathan et al., 1996*).

Charybdotoxin (CTX; *Figure 1B*), a pore-blocking toxin for K⁺ channels, is a 37-residue peptide isolated from the venom of the scorpion *Leiurus quinquestriatus* (*Miller et al., 1985*). Early experiments with CTX inhibition of the BK channel revealed that CTX binds to the extracellular surface of the channel with a 1:1 channel:toxin stoichiometry, that both the open and closed states of the channel are competent for toxin binding, and that electrostatic interactions play an important role in enhancing the toxin's affinity (*Anderson et al., 1988*). Furthermore, CTX affinity was found to be voltage dependent, a property later shown to result from the destabilization of the toxin-channel complex by permeant ions entering from the intracellular side (an effect called 'trans-enhanced dissociation'; *MacKinnon and Miller, 1988*). Ions that were unable to traverse the ion conduction pathway did not elicit trans-enhanced dissociation. These observations led to a hypothesis that CTX physically occludes the ion-conduction pathway, and in doing so brings a positive charge on CTX close to a K⁺ ion-binding site near the extracellular side (*MacKinnon and Miller, 1988*). The positive charge was later identified as Lys27, a residue that is conserved in all members of the CTX-like toxin family (*Figure 2A*; *Park and Miller, 1992*; *Goldstein and Miller, 1993*). Studies with other members of the CTX toxin family, most extensively, Agitoxin2 (AgTx2), supported the conclusion that they function in a manner similar to CTX

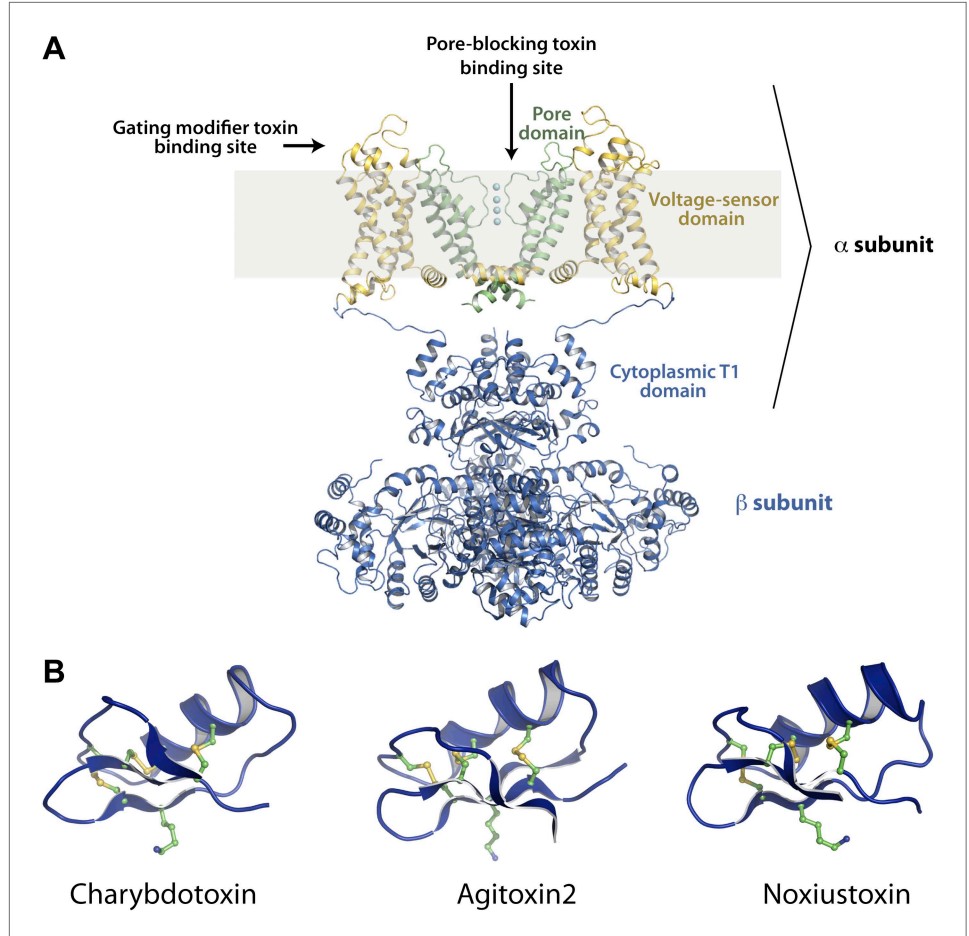

**Figure 1**. Structures of the channel used in this study and of representative scorpion toxins, including the one used in this study. (**A**) Side view showing two diagonal subunits of the paddle chimera channel in complex with the auxiliary β-subunit shown in ribbon trace (PDB ID 2R9R; **Long et al., 2007**). The pore domain of paddle chimera is colored green, and the voltage sensor domain and the linker between the voltage sensor and the pore are colored in yellow. The cytoplasmic T1 domain and the auxiliary β-subunit are shown in blue. The $K^+$ ions in the selectivity filter are shown as cyan spheres. The area corresponding to the membrane is shaded in light gray. The channel forming α-subunit is indicated. Each α-subunit forms a complex with a β-subunit and four such α- and β-heterodimers make up the tetramer. Note that because the voltage sensors arrange around the pore domains in a domain-swapped fashion, the voltage sensor domains and the pore domains shown in the figure belong to different molecules of the tetrameric channel. Sites on the channel for binding the pore-blocking toxins and gating-modifier toxins are shown with arrows. (**B**) The lowest energy NMR structures of representative scorpion toxins with activity on $K_v$ channels—charybdotoxin (CTX; PDB ID 2CRD; **Bontems et al., 1991**), Agitoxin2 (AgTx2; PDB ID 1AGT; **Krezel et al., 1995**) and Noxiustoxin (PDB ID 1SXM; **Dauplais et al., 1995**) are shown in blue ribbon trace. A critical lysine (Lys27 for CTX and AgTx2 and Lys28 for Noxiustoxin) that is conserved in this family of toxins and the conserved cysteines are shown in ball and stick rendition and are colored by atoms.

(**Garcia et al., 1994**; **Krezel et al., 1995**; **Hidalgo and MacKinnon, 1995**; **Ranganathan et al., 1996**). Most notably, conservation of the toxin shape and the functionally important lysine suggested that they all bind with a similar orientation on the $K^+$ channel and inhibit through a common mechanism, whereby a lysine amino group functions as a $K^+$ ion mimic to block the pore (**Miller, 1995**; **Figure 1B**).

Double-mutant cycle studies between toxins (CTX and AgTx2) and the Shaker $K^+$ channel provided numerous pairwise restraints for mapping the extracellular-facing pore surface (**Goldstein et al., 1994**; **Gross et al., 1994**; **Stampe et al., 1994**; **Hidalgo and MacKinnon, 1995**; **Naranjo and Miller, 1996**; **Ranganathan et al., 1996**). NMR-derived models using the KcsA $K^+$ channel also provided valuable

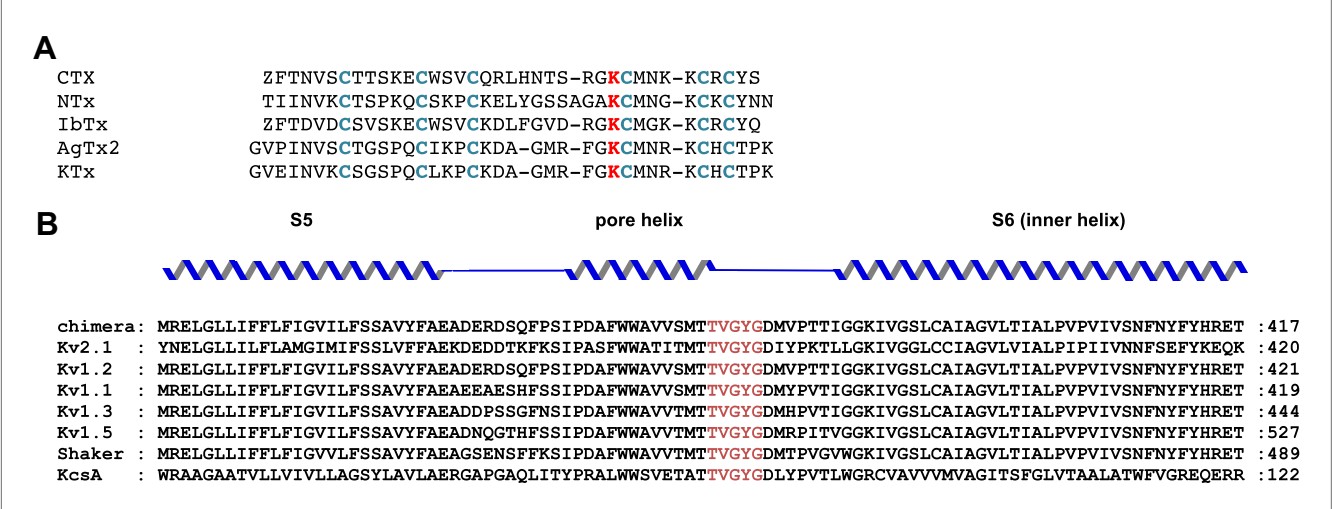

**Figure 2**. Sequence alignments of toxins and pore regions of K+ channels. (**A**) Sequence alignments of five toxins belonging to the CTX family of K+ channel toxins. The conserved cysteines that form disulfide bonds are shown in blue and the conserved lysine that competes with K+ is shown in red. (**B**) Sequence alignment of the pore regions of selected members of the Shaker family of channels and KcsA–paddle chimera, rat K$_v$2.1 (GI:24418849), rat K$_v$1.2 (GI:1235594), human K$_v$1.1 (GI:119395748), human K$_v$ 1.3 (GI:88758565), human K$_v$1.5 (GI:25952087), Shaker K$_v$ (GI:13432103), and KcsA (GI:61226909). The sequence of the selectivity filter is shown in red.

structural data (*Takeuchi et al., 2003*; *Yu et al., 2005*). However, models derived from the double-mutant cycle and NMR data were largely silent as to the influence of toxin on the conducting ions. Here, we have used x-ray crystallography to determine the structure of a complex between CTX and the paddle chimera, a mutant of the K$_v$1.2 K+ channel from rat brain, with particular focus on the influence of toxin on the selectivity filter structure and distribution of ions in the pore (*Figure 2B*; *Alabi et al., 2007*; *Long et al., 2007*).

## Results

### Overall architecture of the toxin-channel complex

Electrophysiological studies of paddle chimera in planar lipid bilayers had revealed that CTX inhibits paddle chimera with high affinity (~20 nM K$_d$; *Tao and MacKinnon, 2008*). We crystallized the complex of paddle chimera with CTX by mixing together separately purified preparations of the channel and the toxin, and setting up cocrystallization trials. The highest resolution data were obtained from the complex of paddle chimera with the selenomethionine derivative of CTX. We used this dataset to solve the structure of the toxin complex of paddle chimera to 2.5 Å resolution. The architecture of the paddle chimera channel typifies the family of eukaryotic K$_v$ channels such as Shaker, with four pore domains together forming the ion-conduction pathway through the membrane and four voltage sensor domains surrounding the pore (*Figure 1A*; *Long et al., 2007*). The voltage-sensors are linked to the cytoplasmic T1 domains that form a cytosolic tetrameric interface. Each channel-forming α-subunit is associated with an accessory β-subunit on the cytoplasmic side. The toxin channel complex shares the same overall architecture, with the fourfold symmetry axis of the channel tetramer coinciding with the fourfold crystallographic symmetry axis. There are two molecules of the α- and β-heterodimeric complex in the asymmetric unit (*Figure 3*). We refer to these as molecule A (top) and molecule B (bottom). Consequently, the symmetry operations generate two distinct tetramers in the lattice. For the toxin-channel complex, an initial omit map without any toxin in the model clearly shows electron density corresponding to the toxin at the pore entryway (*Figure 4A*). However, the electron density for the toxin bound to molecule A, henceforth referred to as toxin A, was clearer and so we used this map for building a model of the toxin-channel complex.

### Building a model for the toxin

The consequences of an asymmetric toxin, binding to a fourfold symmetric channel, are that the toxin can bind to the tetrameric channel in four statistically distinguishable but structurally and energetically

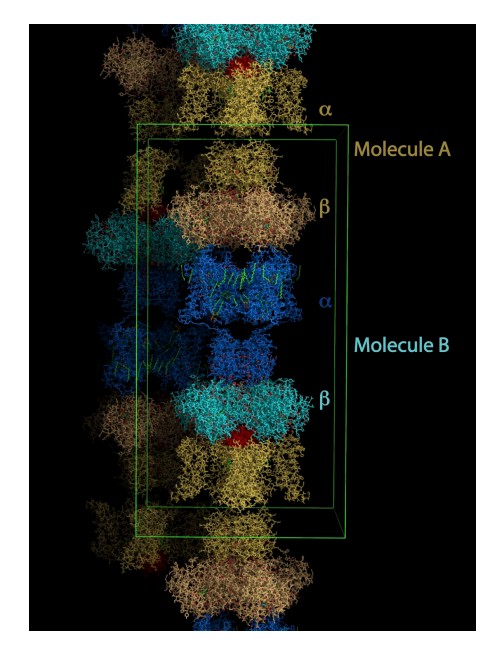

**Figure 3**. Lattice structure of paddle chimera–CTX complex. The asymmetric unit contains two independent molecules of channel forming α-subunits each in complex with an auxiliary β-subunit (see **Figure 1A**). They are called molecule A, shown in stick rendition in different shades of cream; and molecule B, shown in stick rendition in different shades of blue. The toxin bound to molecule A was modeled and is shown in red. The outlines of a unit cell are shown in green.

equivalent orientations. In this case, these orientations are related by the fourfold symmetry axis. Thus, the observed electron density map is a superposition of the electron densities for four such individual orientations of the toxin (**Figure 4A**). Moreover, since in the absence of any external constraints, each orientation is populated with one-fourth occupancy, the electron density for each orientation is inherently weak. Certain secondary structural features in the map were discernible and allowed approximate placement of a CTX molecule, whose structure was determined previously using NMR (**Figures 1B, 4A**; **Bontems et al., 1991**).

It was apparent from the outset that this initial model would require additional data to achieve a reasonable level of accuracy. To obtain the additional data, we incorporated electron-dense marker atoms individually, at several sites on the toxin and purified each heavy atom-modified toxin either by peptide synthesis followed by refolding and purification or by overexpression in *Escherichia coli* and following literature procedures (**Park et al., 1991**). We used three different heavy atom markers—replacement of a disulfide by a diselenide, 4-iodophenylalanine, and selenomethionine (**Figure 4B**). We tested each of these derivatives in a planar bilayer system, and they efficiently blocked the paddle chimera channel (**Figure 4—figure supplement 1**). We then crystallized each toxin derivative with paddle chimera and collected single-wavelength anomalous diffraction data at an appropriate wavelength for each derivative. For each derivative, an anomalous difference electron density map showed four heavy-atom peaks, corresponding to the four orientations of the toxin (**Figure 5A**). We collected datasets from crystals with each derivative at a distinct site on the toxin, a total of three datasets (**Table 1**). We next used our highest resolution dataset (of all the different toxin derivative complexes) and roughly placed the toxin in the omit electron density map (**Figure 4A**). We used this approximate initial placement of the toxin to determine which one of the four symmetry-equivalent peaks in the anomalous difference map (for each marker) corresponded to which orientation of our initial toxin placement. This provided a set of three experimental constraints corresponding to the three individual heavy atom peak positions (for three different markers; **Figure 5B**). The coordinates of the corresponding three sites from the known NMR structure of the toxin (**Bontems et al., 1991**) were used as three reference constraints. We then used RMSD-based superposition to minimize the sum of distances between the peaks in the map and the predicted positions on the toxin (**Figure 5C**; final RMSD 1.6 Å). This procedure yielded a constrained placement of the toxin, which was subsequently refined by rigid-body, coordinate-based B-factor refinement in CNS with manual adjustments, where appropriate (**Figure 5D**).

As noted previously, the density for toxin A in the omit map is better defined than for toxin B, and thus we chose toxin A density for initial placement and for building and refining the model of the toxin. However, when we superimposed the toxin-bound channel A onto channel B using the pore domains of channels A and B for the superposition, we observed a very reasonable model for toxin B, which agrees well with the omit electron density map for toxin B and places the side-chain of Met29 very near to one of the four experimentally observed heavy atom peaks for toxin B in the selenomethionine dataset (**Figure 5—figure supplement 1**). Since channels A and B are independent molecules in the asymmetric unit and only the heavy atom peak positions for toxin A were used for the initial placement of the toxin, this provided further validation of our model and boosted our confidence in the

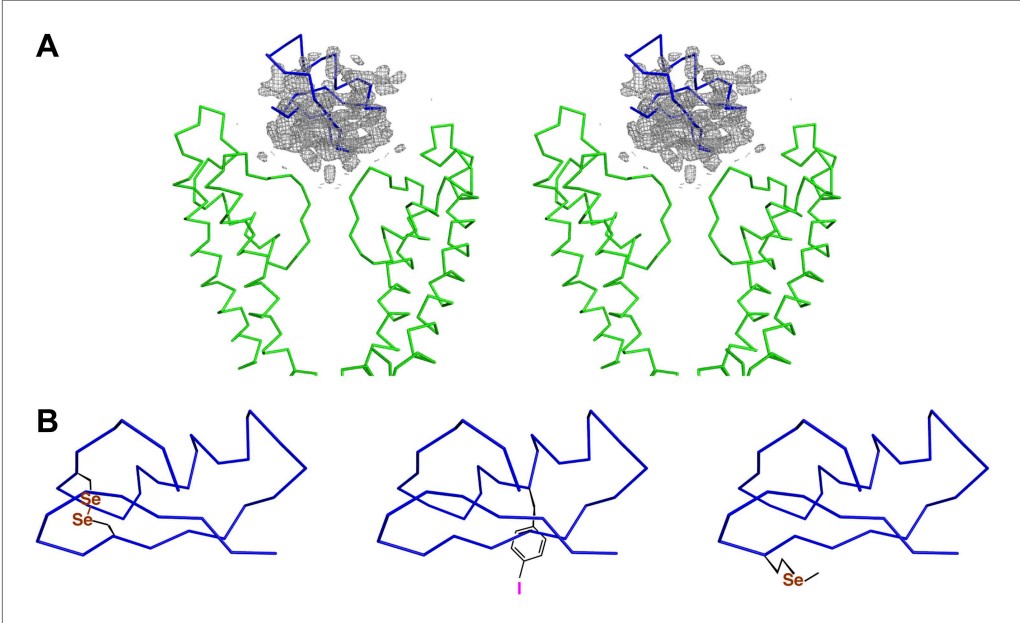

Figure 4. Initial placement of the CTX molecule in the toxin-channel complex and heavy atom derivatives used for subsequently improving the accuracy of placement. (**A**) Stereoview showing the pore domains from two diagonal subunits (molecule A) of paddle chimera (in the toxin-channel complex) in green α carbon trace and a toxin-omit weighted $2F_o - F_c$ electron density map in wire mesh at 0.8 σ contour level. The initial placement of the CTX molecule using the NMR structure (PDB ID 2CRD; *Bontems et al., 1991*) is shown in blue α carbon trace within the omit map. (**B**) Chemical structures of the heavy atom derivatives of CTX used in this study are shown schematically. The NMR structure (PDB ID 2CRD; *Bontems et al., 1991*) is used to depict the rest of the molecule in blue α carbon trace. The heavy atoms are shown in color, purple for iodine and copper for selenium.

The following figure supplements are available for figure 4:

**Figure supplement 1**. Inhibition of paddle chimera by the different heavy atom derivatives of CTX used in this study.

placement of the toxin. We did not include a model for toxin B in our final model since the overall electron density is not as well defined as for toxin A and building toxin B caused a small increase in $R_{free}$.

## Structural features of the toxin-channel complex

Superposition of the channel in the toxin complex onto the channel in the toxin-free structure (*Long et al., 2007*) shows that the channel undergoes no discernible structural changes (*Figure 6A,F*); RMSD 0.33 Å, residues 321–414, main chain atoms, molecule A; RMSD 0.16 Å, residues 321–414, main chain atoms, molecule B. This is consistent with the idea that the toxin fits into the mouth of the channel in a lock and key manner. The oblate-shaped toxin binds asymmetrically to the mouth of the pore domain of the channel such that the wider end is closer to the symmetry axis of the channel than the tapered end (*Figure 6B*). The helical part of the toxin molecule faces away from the channel and the edge of the toxin formed by the residues 25–29 on a β-strand faces toward the channel. The inherent architecture of this class of toxins is such that the three disulfide bonds that hold the folded toxin together are the main buried components in the structure and most of the side chains are displayed on the surface of the toxin. These side chains are in a position to engage into a number of different kinds of interactions with the channel molecule (*Figures 7A,B*). Closer to the fourfold symmetry axis of the channel, the aromatic ring of Tyr36 is positioned to pack simultaneously against Asp375 and Val377 of one subunit and Met29 is able to pack against Asp375 of an adjacent subunit (*Figure 7A,B*). Closer to the periphery, the side chain of Arg25 is within close proximity of Gln353, and the peptide backbone near Thr8-Thr9 is held against Gln353 of another subunit (*Figure 7A,B*). There are also residues that should be involved in long-range electrostatic interactions, that is, Arg25 is within 5.5 Å of Asp359. The guanidium headgroup of Arg25 could also, in principle, engage in electrostatic interactions with an

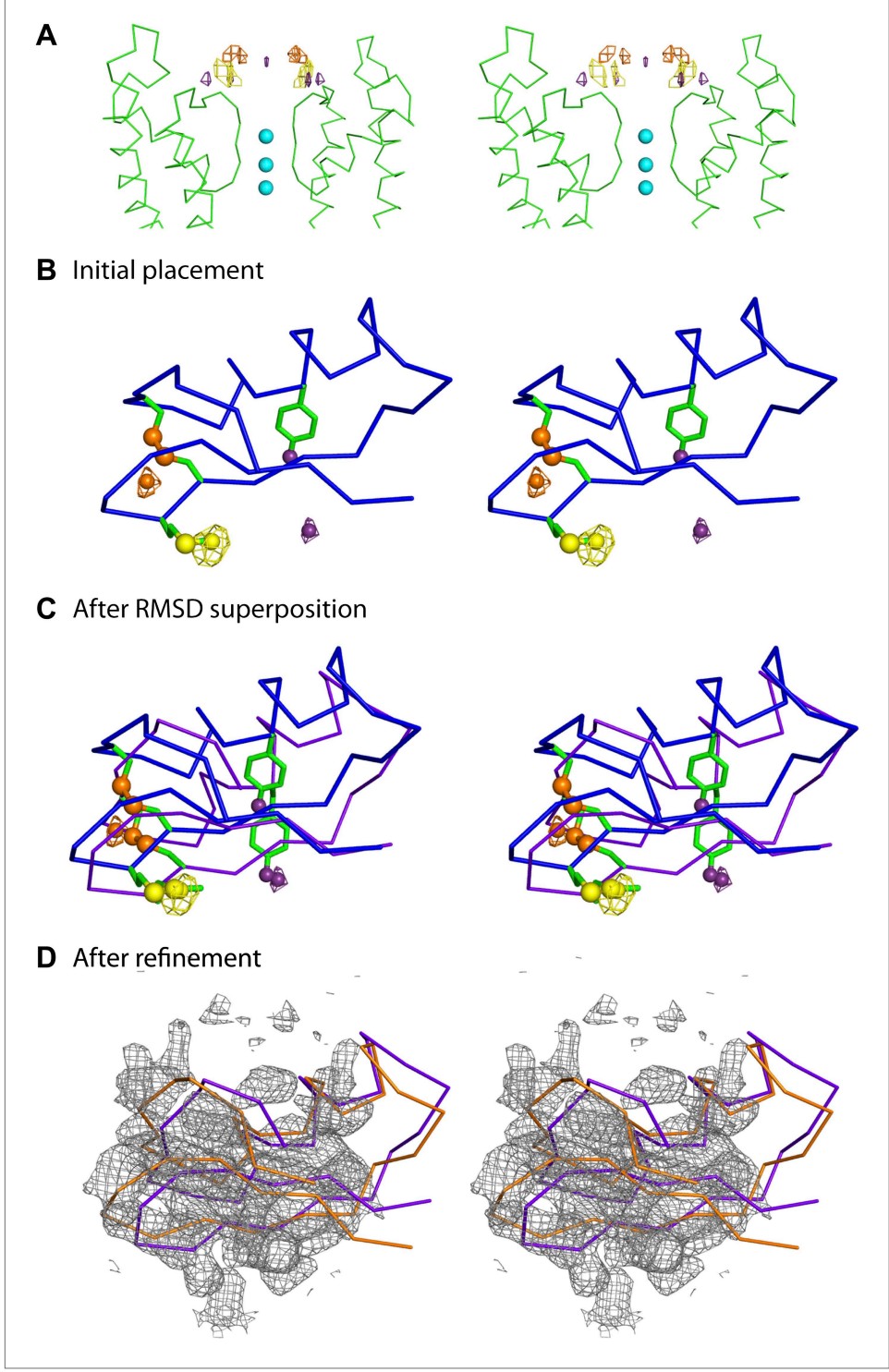

**Figure 5**. Improvement of the initial placement of the toxin. (**A**) Close-up stereoview showing part of the pore domains from two diagonal subunits (molecule A) of paddle chimera. Peak positions from the anomalous electron density maps for the three derivatives (see **Figure 4B**) are shown and colored as follows—purple (4-iodophenylalanine mutant at position 14), orange (diselenide mutant of Cys7-Cys28), and yellow (SeMet mutant at position 29). Note that the four peaks correspond to the four positions of the toxin. In addition, the peak on the symmetry axis in the anomalous electron density map of the 4-iodophenylalanine mutant likely derives from reinforcement of noise peaks that are very close to the symmetry axis. (**B**) Stereoview showing initial placement of CTX in the omit map using the NMR structure (PDB ID 2CRD; **Bontems et al., 1991**), in blue α carbon trace (same as in **Figure 4A**). The heavy

*Figure 5. Continued on next page*

*Figure 5. Continued*

atoms in the depicted orientation are shown as oversized spheres with the corresponding peak positions (the closest of the four shown in *Figure 5A*) in the anomalous electron density maps in wire mesh. Color-coding of the maps are the same as in *Figure 5A*. A dummy atom has been placed to indicate the position of each peak. (**C**) The toxin molecule is shown in purple α carbon trace, after RMSD superposition of the heavy atom positions in the structure onto experimental peak positions as illustrated by the dummy atoms in *Figure 5B*. The initial placement of the toxin as shown in *Figure 5B* is also shown in blue. (**D**) The final refined model of the toxin after crystallographic refinement (please see text and 'Materials and methods') is shown in orange α carbon trace together with the model after initial RMSD superposition in purple and a weighted $2F_o − F_c$ electron density map in wire mesh at 1σ contour level.

The following figure supplements are available for figure 5:

**Figure supplement 1**. Validation of the model for the toxin.

ordered lipid that is present in the structure. Although the density of the lipid headgroup is not clear in this case, it is likely that the headgroup will be placed close enough for the Arg25 to make electrostatic contact with it. Arg34 is another residue that approaches the channel closely enough to make electrostatic interactions, that is, with the carbonyl oxygen of Asp375. Arg34 is also within H-bonding distance of Gln353 and within long-range electrostatic contact of Asp375. Asn30 is another residue that appears to approach close to Asp375 to enable a weak H-bonding interaction.

The most striking aspect of the toxin-channel complex concerns the distribution of $K^+$ ions in the selectivity filter of the channel. In the structure of the paddle chimera, as well as other high-resolution $K^+$ channel structures, the selectivity filter contains four distinct ion-binding sites or positions, S1 through S4, S1 being the most extracellular (*Figure 6C*; *Zhou et al., 2001*; *Long et al., 2007*; *Nishida et al., 2007*). From analyses of high-resolution diffraction data on KcsA, the prototypical $K^+$ channel pore, it was inferred that during conduction, these ion-binding sites are occupied alternately in what are referred to as 1,3 and 2,4 configurations (*Morais-Cabral et al., 2001*). In accordance, the occupancy of each site was experimentally determined to be roughly 0.5 (*Zhou and MacKinnon, 2003*). In the toxin-channel complex, only sites S2 through S4 have discernible electron density (*Figure 6D*). The top ion-binding site appears empty. It is important to note here that this observation holds strictly true for both channel molecules in the asymmetric unit. We have several datasets for all the different derivative toxin complexes, and this observation holds true for all of them as well, in both channel molecules A and B, in the asymmetric unit.

Why is the distribution of ions dramatically different in the toxin-bound complex? The toxin binds at the mouth of the pore positioned in such a manner to project the side chain of Lys27 straight into the pore, allowing the amino group to approach the top of site S1 (*Figure 6E* and *Figure 6—figure supplement 1*). Thus, Lys27 is within the range to make hydrogen-bonding interactions with all four carbonyl oxygen atoms that would otherwise constitute the top-half layer of coordinating ligands for a $K^+$ at site S1 (*Figure 6E*; *Zhou et al., 2001*). We suspect therefore two reasons why $K^+$ is disfavored at site S1. First, there is electrostatic repulsion from the closely placed positively charged amino group, and second, carbonyl oxygen atoms that would otherwise constitute half of the coordination are not fully available for coordination with a $K^+$ at site S1. The structure thus offers a simple rationale for the altered distribution of ions in the selectivity filter of the toxin-bound channel.

## Structure of the toxin-channel complex in $Cs^+$ and structure with a Lys27 mutant of CTX

The altered ion distribution informs us that the toxin interacts with ions inside the pore. This interaction is compatible with the electrophysiological observation that intracellular ions destabilize extracellular toxin—the trans-enhanced dissociation effect (*MacKinnon and Miller, 1988*; *Park and Miller, 1992*). To further correlate the ion distribution in the crystal with trans-enhanced dissociation in electrophysiology experiments, we crystallized the channel with a mutant toxin, Lys27Met. Miller and coworkers had shown that Lys27 mutants reduce toxin affinity and abolish trans-enhanced dissociation (*Park and Miller, 1992*; *Goldstein et al., 1994*). Lys27Met CTX indeed inhibits the paddle chimera channel with reduced affinity (~630 nM) but still forms a complex at the high concentrations of channel (~20 μM) and toxin (60–80 μM) present in crystallization trials (*Figure 8A,B*). Electron density is present at S1 in an ion omit map, compatible with the presence of an ion, although the density at S1 is weaker

**Table 1.** Crystallographic data and refinement statistics table

| Data collection | | | |
|---|---|---|---|
| Dataset | Paddle chimera–SeMet CTX | Paddle chimera–Lys27Met CTX | Paddle chimera–CTX in CsCl |
| Space group | P42$_1$2 | P42$_1$2 | P42$_1$2 |
| Cell constants (Å) | a = b = 144.200; c = 283.608 | a = b = 144.842; c = 283.938 | a = b = 145.434; c = 285.591 |
| | α = β = γ = 90° | α = β = γ = 90° | α = β = γ = 90° |
| Source | BNL X29 | BNL X29 | BNL X29 |
| Wavelength (Å) | 0.9791 | 1.075 | 1.075 |
| Resolution (Å) | 2.5 | 2.54 | 2.56 |
| Unique reflections | 99,907 | 97,913 | 98,762 |
| $\langle I \rangle / \langle \sigma I \rangle$* | 29.4(2.59–2.50/1.9) | 18.4 (2.58–2.54/2.2) | 13 (2.60–2.56/1.04) |
| Redundancy* | 6.9 (2.59–2.50/2.3) | 7.3 (2.58–2.54/6.0) | 7.1 (2.60–2.56/5.3) |
| Completeness (%)* | 96.1 (2.59–2.50/71.1) | 96.7 (2.58–2.54/32.8) | 99.3 (2.60–2.56/87.6) |
| R$_{merge}$ (%)* | 5.9 (2.59–2.50/39.8) | 8.6 (2.58–2.54/73.2) | 9.5 (2.60–2.56/>100) |
| **Model Refinement** | | | |
| Resolution (Å) | 50–2.5 | 50–2.54 | 50–2.56 |
| Reflections (free set) | 96,658 (4605) | 93,308 (4406) | 91,104 (4292) |
| R$_{work}$/R$_{free}$ (%) | 21.1/23.6 | 21.0/23.4 | 23.7/26.2 |
| RMSD bond lengths (Å) | 0.006 | 0.007 | 0.007 |
| RMSD bond angles (°) | 1.09 | 1.249 | 1.299 |
| Mean B-factor (Å$^2$) | 73.09 | 75.07 | 72.11 |
| Ramachandran plot | | | |
| Allowed (%) | 99.5 | 99.4 | 99.4 |
| Disallowed (%) | 0.5 | 0.6 | 0.6 |

*Numbers in parentheses represent the resolution range of the highest resolution shell followed by the value of the parameter for the highest resolution shell.

relative to the ion densities at S2–S4, as if the occupancy at S1 is reduced (*Figure 8C*). Thus, it appears that the ability of toxin to interact with ions in the pore (and thus alter the ion distribution) is directly connected to the ability of ions from the intracellular solution to destabilize toxin on the extracellular side through interaction with Lys27. A simple mechanistic explanation could be that K$^+$ and toxin—via Lys27—compete for stabilizing interactions at S1 in the selectivity filter.

The above mechanistic proposal provides motive to wonder what happens when K$^+$ is replaced with Cs$^+$, because Cs$^+$ in crystal structures of K$^+$ channels binds at only three sites and with unusually high occupancy at S1 (*Zhou and MacKinnon, 2003*). A CTX complex with the paddle chimera channel in the presence of Cs$^+$ shows that toxin is bound but that Cs$^+$ adopts its expected distribution (i.e., similar to its distribution in KcsA in the absence of toxin) with an ion at S1 (*Figure 8D,E*). It is difficult to tell where in the electron density map the amino group of Lys27 resides, but it is clear that Cs$^+$ competes effectively for site S1, despite the presence of CTX. The mechanism of competition put forth above predicts that CTX should not bind with high affinity in the presence of Cs$^+$. As shown, this prediction holds: CTX inhibits with a nearly 10-fold reduced affinity in the presence of Cs$^+$ compared to K$^+$ (*Figure 8A*).

## Discussion

We report here the first x-ray structure of a K$^+$ channel bound to a toxin. Many of the original studies on pore-blocking toxins for K$^+$ channels were carried out with eukaryotic voltage-gated K$^+$ channels, such as Shaker and K$_v$1.3, closely related in sequence to the mutant version of the eukaryotic K$_v$1.2 channel

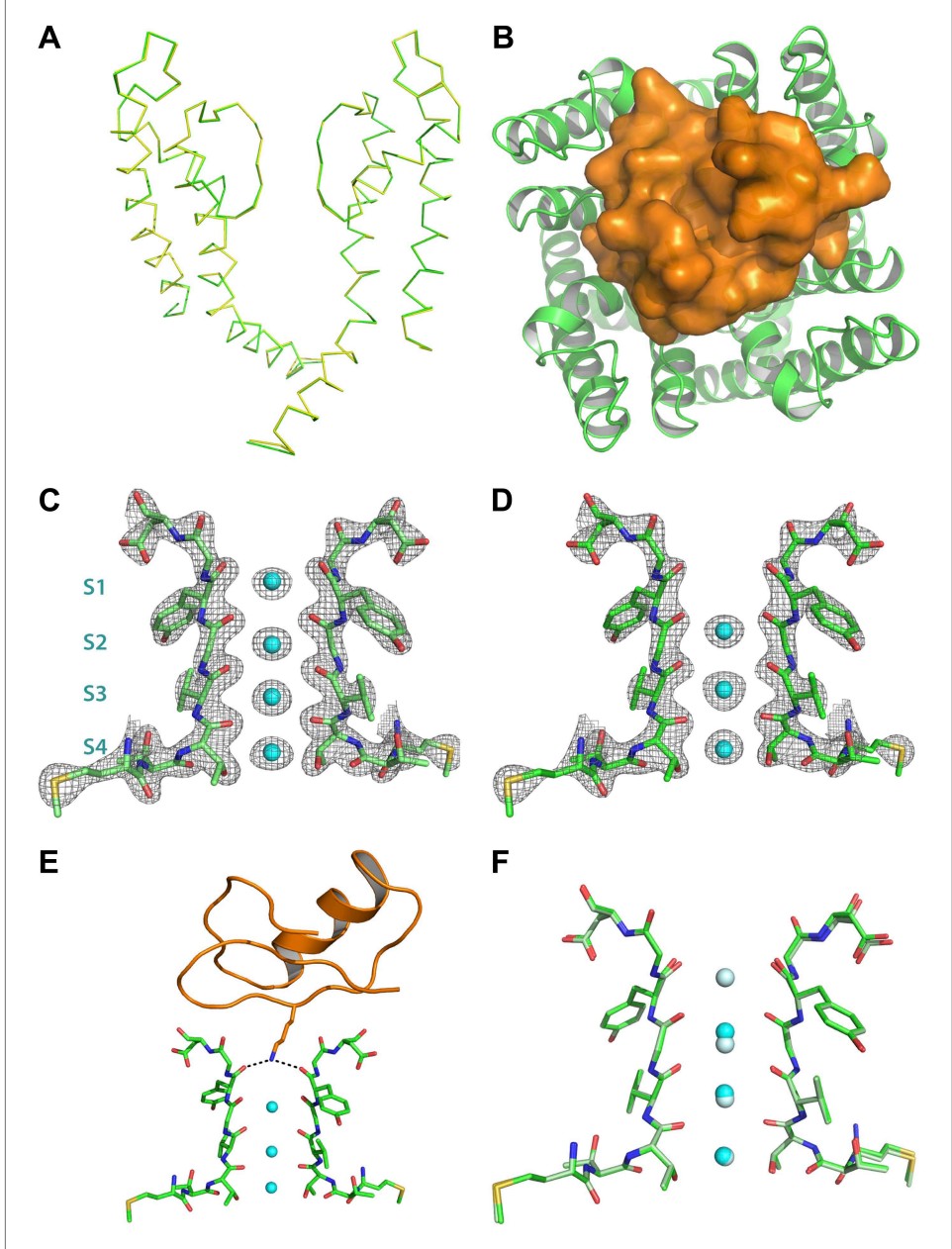

**Figure 6**. Structure of the toxin-channel complex. (**A**) Side view showing pore domains from two diagonal subunits from paddle chimera (yellow; PDB ID 2 R9R; **Long et al., 2007**) and the toxin-channel complex (green) in α carbon trace. They have been superimposed by RMSD superposition of the main chain atoms from residues Met321-Thr414. (**B**) The tetrameric pore domain in the toxin-channel complex is shown in green ribbon trace from an extracellular view looking into the molecule. The bound CTX is shown in surface rendition in orange. (**C**) Side view of the selectivity filter (two diagonal subunits, molecule B) from the paddle chimera structure shown in stick rendition with the K$^+$ ions shown as cyan spheres. Sites S1 through S4 in the selectivity filter are labeled (labels on left side) in cyan. A weighted 2F$_o$ − F$_c$ electron density map contoured at 3σ is shown in wire mesh. (**D**) Side view of the selectivity filter (two diagonal subunits, molecule B) from the toxin-channel complex is shown in stick rendition with the K$^+$ ions shown as cyan spheres. A weighted 2F$_o$ − F$_c$ electron density map contoured at 3σ is shown in wire mesh. (**E**) The selectivity filter from the toxin-channel complex is shown in stick rendition with the K$^+$ ions shown as cyan spheres. Also shown is the bound CTX molecule in orange ribbon trace and the side chain of the Lys27 residue in stick rendition. Close contact between the amino headgroup and the carbonyl oxygen in the selectivity filter are shown in dotted lines. (**F**) RMSD superposed structures of the selectivity filter regions (same

*Figure 6. Continued on next page*

*Figure 6. Continued*

regions as shown in ***Figures 6C,D***) of the paddle chimera (pale green) and the toxin-channel complex (green) shown in stick rendition. The superposition was done using the main chain atoms from residues Met321-Thr414. The K⁺ ions in the paddle chimera structure are shown in light blue and those in the toxin-channel complex in cyan.

The following figure supplements are available for figure 6:

**Figure supplement 1**. Electron density for the side chain of Lys27 of CTX in the paddle chimera–CTX complex.

that we employed in our structural studies (***MacKinnon and Miller, 1989***; ***MacKinnon, 1991***; ***Goldstein and Miller, 1992***; ***Stampe et al., 1992***, ***1994***; ***Goldstein et al., 1994***; ***Gross et al., 1994***; ***Aiyar et al., 1995***, ***1996***; ***Hidalgo and MacKinnon, 1995***; ***Gross and MacKinnon, 1996***; ***Naini and Miller, 1996***; ***Naranjo and Miller, 1996***; ***Ranganathan et al., 1996***; ***MacKinnon et al., 1998***). Our first major finding is that we do not see discernible changes in the structure of the channel between the toxin-bound and the toxin-free paddle chimera structures (***Figure 6A,F***). This, we note, is in contrast to the solid-state NMR 'structure' of a KcsA mutant with kaliotoxin, where such rearrangements in the channel were proposed (***Lange et al., 2006***). In this NMR study, chemical shift changes induced by toxin were

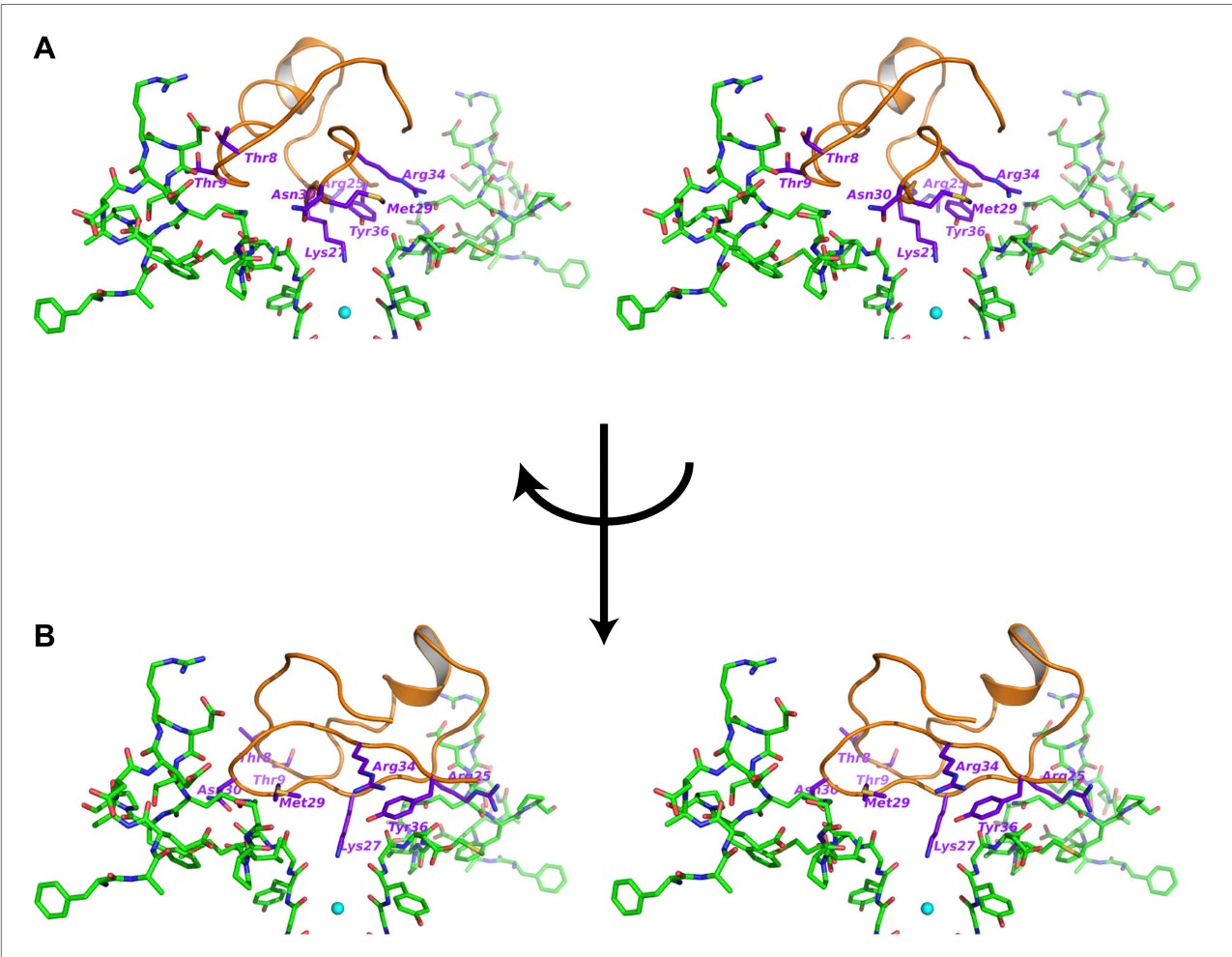

**Figure 7**. Interactions between the bound CTX and the channel. (**A**) Stereoview showing the CTX receptor region of paddle chimera from the toxin-channel complex shown in stick rendition together with the top parts of the selectivity filter. The bound CTX is shown in orange ribbon trace. Side chains of selected residues of the toxin are shown in purple and are labeled. (**B**) Shows a view orthogonal to that in (**A**). Consequently the receptor regions shown in (**B**) are from the two diagonal subunits of paddle chimera that are not shown in (**A**).

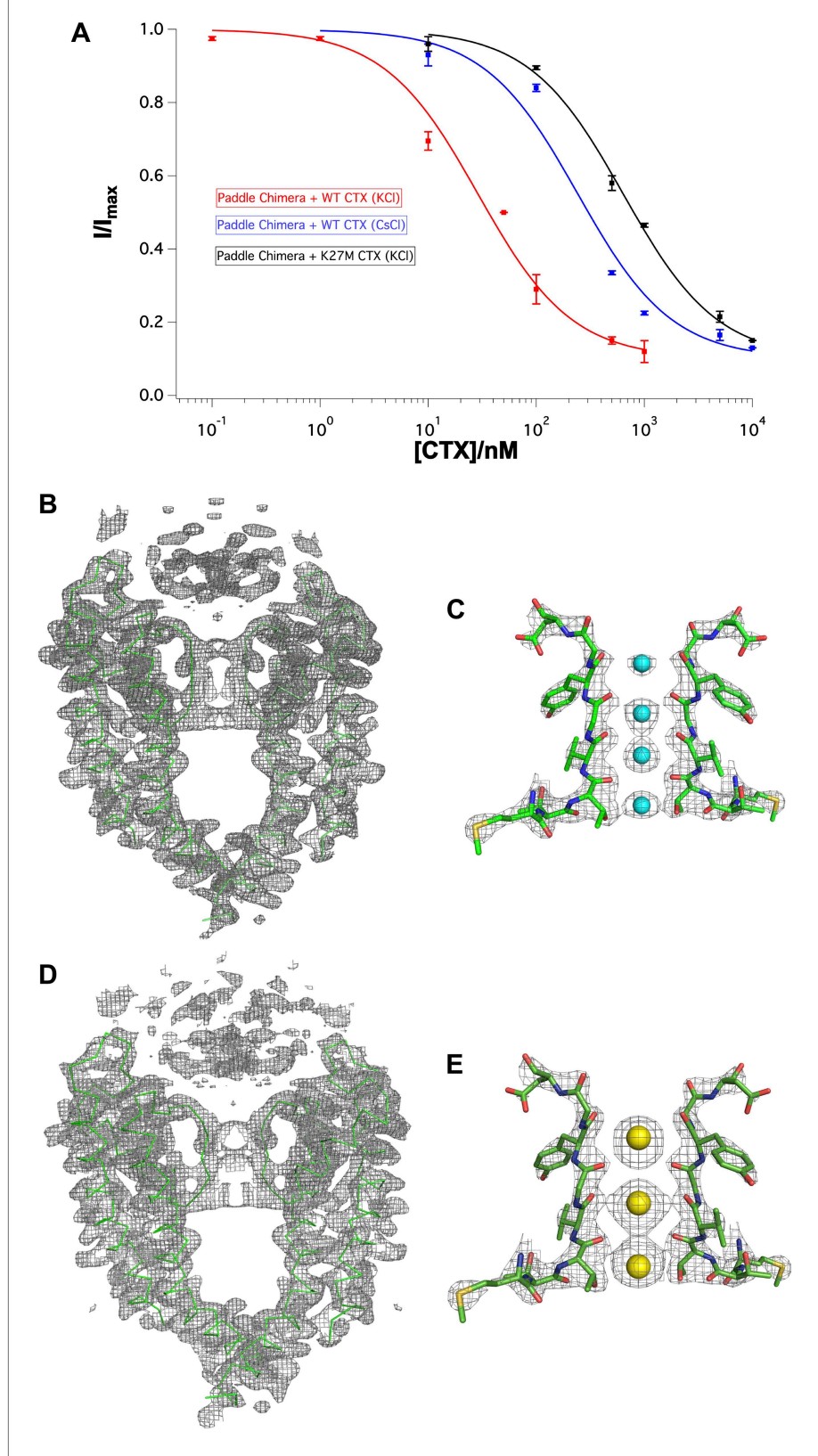

**Figure 8**. Effects of Lys27Met mutation of CTX and Cs⁺ separately on toxin-channel interactions. (**A**) CTX inhibition. The fraction of unblocked current (I/Imax, mean ± SEM or range of mean; n = 2–4) is graphed as a function of CTX concentration (in nM) and fit to the equation I/Imax = 0.1 + 0.9 × (1 + [CTX]/$K_d$)⁻¹. Voltage pulses: holding −110 mV, *Figure 8. Continued on next page*

*Figure 8. Continued*

depolarized to +110 mV, followed by a step back to −110 mV. Paddle chimera–wild-type CTX with KCl on both sides is shown in red, paddle chimera–wild-type CTX with CsCl on both sides is shown in blue and paddle chimera–Lys27Met CTX with KCl on both sides is shown in black. The modified equation accounts for approximately 10% current that is due largely to the channels facing the other side and are not blocked by toxin. (**B**) The pore domains from two diagonal subunits of paddle chimera (molecule A) in the Lys27MetCTX–paddle chimera complex are shown in green α carbon trace and a toxin-omit weighted $2F_o − F_c$ electron density map is shown in wire mesh at 1σ contour level. (**C**) Side view of the selectivity filter (two diagonal subunits; molecule B) of the Lys27Met CTX–paddle chimera complex shown in stick rendition with the $K^+$ ions shown as cyan spheres. An ion-omit weighted $2F_o − F_c$ electron density map is shown in wire mesh at 3.2σ contour level. (**D**) The pore domains from two diagonal subunits of paddle chimera (molecule A) in CTX–paddle chimera complex in CsCl, are shown in green α carbon trace and a toxin-omit weighted $2F_o − F_c$ electron density map is shown in wire mesh at 0.8σ contour level. (**E**) Side view of the selectivity filter (two diagonal subunits; molecule B) of the CTX–paddle chimera complex in CsCl shown in stick rendition with the $Cs^+$ ions shown as yellow spheres. A weighted $2F_o − F_c$ electron density map is shown in wire mesh at 3σ contour level.

interpreted as resulting from dihedral angle changes (i.e., structural); however, such chemical shift changes could have other origins (i.e., electrostatic). Moreover, no channel-toxin distance restraints were included in the determination of this solid-state NMR 'structure' (*Lange et al., 2006*). In the crystal structure presented here, the good precomplex complementarity between the shape of the channel entryway and the shape of the toxin (i.e., the absence of channel structural change upon toxin binding) suggests a possible explanation for two prominent features of pore-blocking toxins. First, toxins can bind with relatively high affinity to their target channels because binding free energy is not 'spent' bringing about a protein conformational change. And second, single mutations in the 'toxin receptor' region of the channel can drastically alter the affinity for the channel by disrupting the good fit (*Goldstein et al., 1994*; *Garcia et al., 1997*).

The relatively static architecture of the $K^+$ channel pore entryway undoubtedly reflects the requirement of a well-ordered selectivity filter structure to select $K^+$ ions. In hindsight, this static toxin receptor on $K_v$ channels lends credence to the idea that was originally proposed for using the pore-blocking toxins as 'molecular slide calipers' for gauging, at the resolution of mutagenesis, the shape of the protein surface on the extracellular part of the pore domain (*Goldstein et al., 1994*; *Stampe et al., 1994*; *Hidalgo and MacKinnon, 1995*; *Ranganathan et al., 1996*). A large body of data has emerged from these studies on mutagenesis-based electrophysiological measurements of toxin-channel interactions in the $K_v$ channel family, mainly using the toxin-channel pairs Shaker–CTX, Shaker–AgTx2, and to a lesser extent, $K_v$1.3–CTX (*Goldstein et al., 1994*; *Stocker and Miller, 1994*; *Hidalgo and MacKinnon, 1995*; *Aiyar et al., 1995*; *Naini and Miller, 1996*; *Naranjo and Miller, 1996*; *Ranganathan et al., 1996*; *Rauer et al., 2000*). We have mapped these data onto our structure. Among these measurements, the data obtained with mutant cycle analyses are more reliable in gauging residue proximities on either side of the toxin-channel interface. The mapping has been done in the following manner: for the channel, we have used a sequence alignment (*Figure 2B*) to match the corresponding residue in Shaker or $K_v$1.3 onto paddle chimera. Data from the studies using CTX are shown in *Figure 9A* and those using AgTx2 in *Figure 9B*. For AgTx2, we used the known structure of AgTx2 and the guidelines in *Krezel et al. (1995)* to superimpose AgTx2 onto CTX in the structure of the paddle chimera–CTX complex. A few data points were derived from lysine scanning, which alters the length of the wild-type residue appreciably. For the lysine mutations, for the sake of representation, the corresponding residue in paddle chimera was mutated in silico (in Coot) to lysine, and the rotamer of lysine with the least distance between the corresponding toxin residue was chosen. Our structure is overall in excellent agreement with not only the CTX data but also with the AgTx2 data.

We wish to discuss two cases of proximity deduced by the mutant cycle analyses and how the structure reveals atomic insights into them. From analyzing mutants of AgTx2 and Shaker, MacKinnon and Ranganathan derived a coupling energy of >3 kT for the Gly10Val(AgTx2)–Phe425Gly(Shaker) pair and ~1.5 kT for the Gly10Val(AgTx2)–Thr449Cys(Shaker) pair (*Ranganathan et al., 1996*). Phe425 and Thr449 in Shaker map onto Gln353 and Val377 in paddle chimera (*Figure 2B*). An inspection of the structure reveals clearly that Gly10 is close to Gln353 (*Figure 10A*), which is consistent with the high coupling energy. Intriguingly, Val377 is not within the first layer of surrounding residues contacted

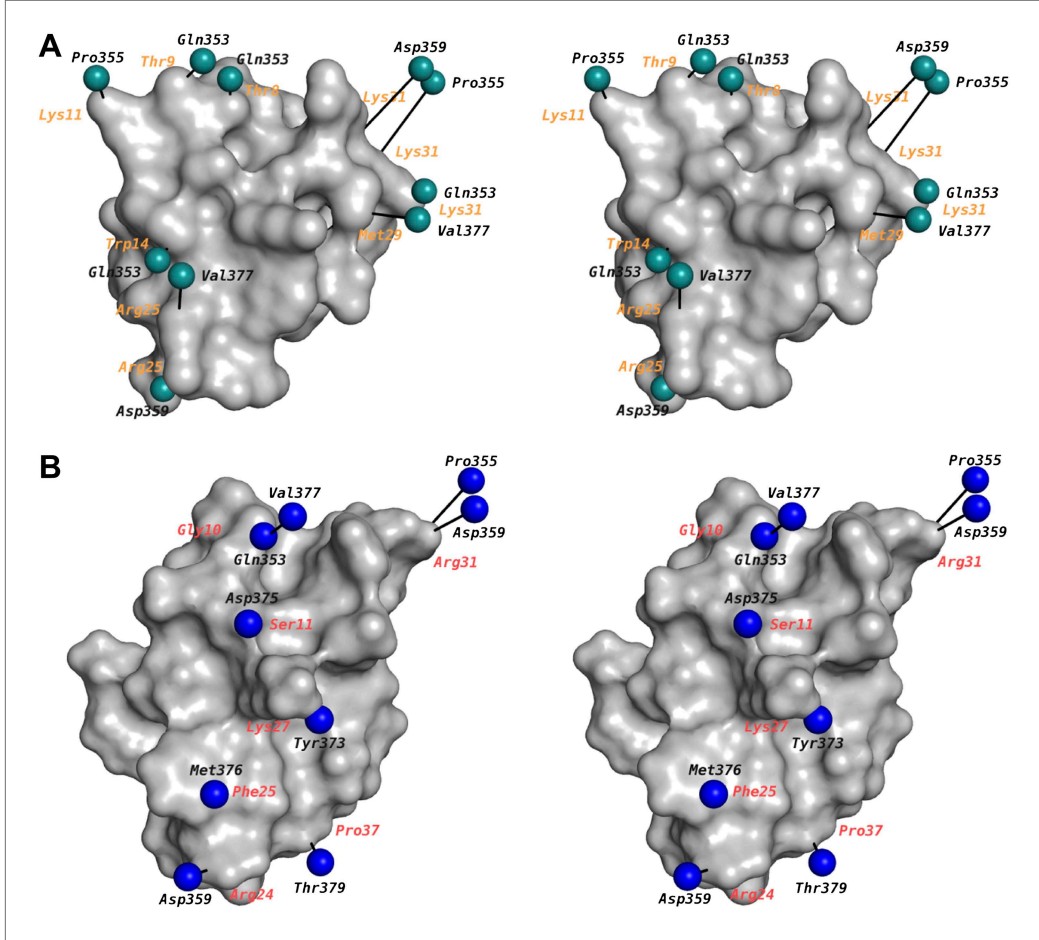

**Figure 9**. Mapping of toxin-channel interactions reported in literature on the structure of paddle chimera–CTX complex. (**A**) Shown is a stereoview of the bound CTX in surface rendition in the CTX–paddle chimera structure from an intracellular perspective viewing down the fourfold symmetry axis of the channel (channel not shown). Residues in the channel that have been reported in the literature to be proximal to the toxin are represented as green spheres taking the coordinates of the closest atom from the structure of the CTX–paddle chimera complex (see text). They are connected to the corresponding residues in the toxin with a black line. The corresponding residues in CTX are indicated in orange. Note that certain residues on the channel have been reported to be proximal to multiple residues on the toxin and thus they are represented more than once in the map. This figure shows data derived with CTX. (**B**) Shown are the residues in AgTx2 reported in the literature to be proximal to the channel, in the same format as (**A**). In order to depict AgTx2, it was superimposed on CTX in the paddle chimera–CTX complex using the published NMR structure of AgTx2 (PDB ID 1AGT; *Krezel et al., 1995*) and using the guidelines in *Krezel et al. (1995)*. AgTx2 is shown in surface rendition and the residues on the channel proximal to AgTx2 are shown as blue spheres. The corresponding residues in AgTx2 are indicated in red.

by Gly10. However, Val377 is within close proximity to Gln353 such that a mutation of Gly10 to valine would incur a clash of Gln353 with Val377 (*Figure 10A*). It is worthwhile noting that pairwise mutant cycle analysis per se does not distinguish between direct interactions and such interactions mediated by a third residue. However, the relative magnitude of the coupling energies, in hindsight, is consistent with such a mode of interaction.

Another case worth highlighting is the coupling energy, ~1.5 kT, between the Lys27Met(AgTx2)–Tyr445Phe(Shaker) pair. Tyr445 in Shaker maps to Tyr373 in paddle chimera (*Figure 2B*; *Ranganathan et al., 1996*). Since mutation of Tyr to Phe incurs only a loss of the hydroxyl group, one likely interpretation of this would have been that Lys27 contacts the hydroxyl group of Tyr445. Intriguingly, Lys27 only contacts the backbone carbonyl oxygen of Tyr445, a point farthest from the hydroxyl group

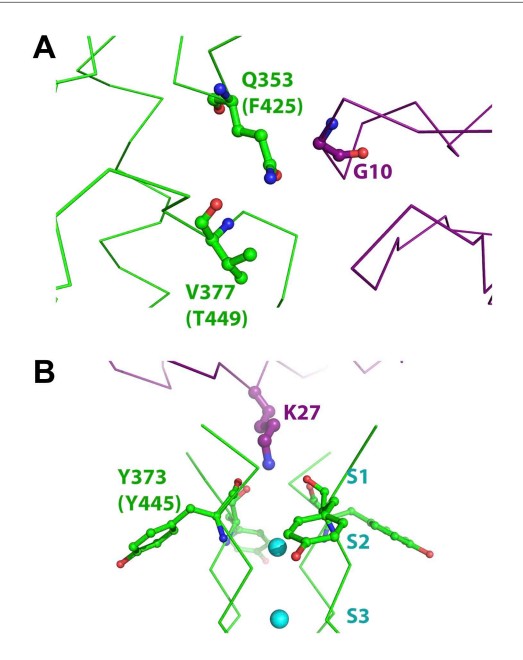

**Figure 10**. Mutant cycle data from *Ranganathan et al. (1996)* mapped onto the structure of paddle chimera–CTX complex. The toxin molecule shown in purple is AgTx2, which was superimposed onto CTX in the structure of the paddle chimera–CTX complex using the NMR structure of AgTx2 (PDB ID 1AGT; *Krezel et al., 1995*). Guidelines in *Krezel et al. (1995)* were used for the superposition. (**A**) Close-up view of AgTx2 showing Glycine 10 in ball and stick rendition and the rest of the toxin in α carbon trace. The channel subunit that is most proximal to Gly10 is shown in green α carbon trace. Shown in ball and stick are residues Gln353 and Val377 in paddle chimera, that correspond to Phe425 and Thr449, respectively, in Shaker using a sequence alignment (*Figure 2B*). Ranganathan and MacKinnon reported that Gly10 is coupled to Phe425 and Thr449 in Shaker. (**B**) Close-up view of AgTx2 showing the side-chain of Lys27 in ball and stick rendition and the rest of the toxin in α carbon trace. The nearby regions of the selectivity filter from all four subunits of the channel are shown in green α carbon trace. Shown in ball and stick is Tyr373 in all four subunits of paddle chimera, which map onto Tyr445 in Shaker using a sequence alignment (*Figure 2B*). Ranganthan and MacKinnon reported Tyr445 to be coupled to Lys27, and this was dependent on the concentration of K+.

(*Figure 10B*). However, since this is a tightly packed part of the protein structure with the hydroxyl group being a part of an interaction network, mutation of the hydroxyl is felt at the backbone carbonyl by the toxin. MacKinnon and Ranganathan also found that the interaction between these two residues is dependent on the K+ concentration. This is also consistent with Lys27 inserting into the selectivity filter and displacing K+ from site S1.

The structure also shows the chemical rationale behind an experimental observation that is decades old concerning the role of a highly conserved lysine in voltage-dependent block by CTX and other pore-blocking toxins in this family. In studies with the BK channel, *MacKinnon and Miller (1988)* first observed that permeant ions coming through the channel from the intracellular side enhance the dissociation of a toxin bound to its receptor on the extracellular side. This trans-enhanced dissociation effect of permeant ions on CTX block of K+ channels was subsequently confirmed for Shaker as well (*Goldstein and Miller, 1993*). In order to explain this phenomenon, MacKinnon and Miller put forward the hypothesis that the toxin binds at a site close to the ion permeation pathway and places a positive charge close to one of the ion-binding sites in the channel. Additionally, it was also observed that this voltage-dependent and permeant ion–dependent block was nearly completely abolished when Lys27 was mutated to a neutral Asn or Gln residue (*Park and Miller, 1992*). Mutation of no other residue on the toxin had the same effect. This implied that the electrostatic interaction between the K+ in the ion-binding site and the toxin was wholly mediated by this single Lys residue, and the structure of the toxin-paddle chimera complex shows exactly why that is so. In the structure of the WT CTX-K+ complex, the top ion-binding site S1 has no density corresponding to K+ ions (*Figure 6C*) and the positively charged amino group is positioned to form contacts with S1 instead (i.e., making hydrogen-bonding interactions with the carbonyl oxygen atoms comprising the top layer of coordinating ligands at S1; *Figure 6D*). From the structure, it is clear that only the side chain of Lys27 approaches close enough to the S1 site to exert such an effect (*Figure 6—figure supplement 1*). It is worth noting here that the long linear alkyl chain and the tetrahedral disposition of hydrogens around the terminal nitrogen of Lysine make it uniquely and chemically suited for making maximal contacts with the top layer of coordinating carbonyl oxygen atoms at S1. Because this region is buried within the protein, this is likely to be a highly stabilizing interaction. Even a conservative mutation of Lys27 to Arginine causes ~1000-fold destabilization of the toxin-channel complex in Shaker (*Goldstein et al., 1994*).

The CTX family of K⁺ channel toxins has remarkable diversity in sequence and binding affinities to specific channel subtypes. It is interesting to note, in light of the present structure, the basic mechanistic principles underlying the mode of action of these toxins. In a strikingly simple but effective strategy, the toxins target a functional aspect common to all the K⁺ channels—ion conduction. At the structural level, this is executed by presenting the amino group of Lys27, a highly conserved residue (*Figure 2A* and *Figure 6—figure supplement 1*), to the top ion-binding site in the selectivity filter. This acts as a tethered surrogate cation and effectively plugs the ion-conduction pathway. How is this surrogate cation brought to this site in the first place? The structures of these miniproteins are held together by rigid disulfide bonds, that are highly conserved in this and other families of small peptide-based toxins. This provides a relatively rigid scaffold. Through evolutionary sampling of intervening less-conserved residues, individual toxins have gained the ability to engage in specific interactions by long-range electrostatic interactions and a few subtype-specific close contacts, thereby ensuring efficient channel block. This is a remarkable example of combinatorial diversity in nature with the constraints of a rigid scaffold and a conserved mechanism to target one of the most important classes of sensory molecules in biology. Scorpion toxins have evolved to fit like a lock and key into the pore entryway of potassium channels, and disrupt ion conduction through presentation of a lysine amino group that competes with potassium in the selectivity filter.

## Materials and methods

### Molecular biology

#### Mutagenesis

All the mutants were generated using the QuikChange site-directed mutagenesis kit (Agilent, Santa clara, USA) and incorporation of the mutation(s) was verified by sequencing.

### Protein expression and purification

#### Expression and purification of K$_v$2.1 paddle–K$_v$1.2 chimera channel

K$_v$2.1 paddle–K$_v$1.2 chimera channel was expressed and purified as described with slight modifications (*Long et al., 2007*). In brief, the channel was coexpressed with the rat β2-core gene in *Pichia pastoris*, extracted with DDM, and purified on a cobalt affinity column followed by gel filtration on a Superdex-200 column using a 1:1 mixture of Cymal6 and Cymal7 detergents. The paddle chimera-β2 complex thus purified was subsequently concentrated to 8–9 mg/ml (Centricon-100; Millipore, Billerica, MA).

#### Expression and purification of wild-type CTX, Ly27Met CTX, and selenomethionine derivative of CTX

Wild-type and Lys27Met CTX were expressed in BL21(DE3) *E. coli* as a T7 gene 9 fusion protein and purified as described (*Park et al., 1991*). Trypsin was used to cleave the inhibitor from the carrier at the Factor X$_a$ site. The selenomethionine derivative was made using the same method except for modifications that were made in the cell growth conditions to prepare the selenomethionine derivative using the methionine biosynthesis inhibition method as described (*van Duyne et al., 1993*).

### Synthesis and refolding of 4-iodophenylalanine and diselenide mutants of CTX

#### Materials

Fmoc protected amino acids, Wang resins, and 2-(1H-benzotriazol-1-yl)-1,1,3,3-tetramethyluronium hexafluorophosphate (HBTU) were purchased from Novabiochem (Switzerland). Fmoc-4-iodophenylalanine, Fmoc-(Se-p-methoxybenzyl)–selenocysteine, and Pyroglutamic acid were purchased from Anaspec (Freemont, CA). HPLC grade MeCN, dichloromethane (DCM), dimethylformamide (DMF), MeOH, and *N*-methylpyrrolidinone (NMP) were purchased from Fischer Scientific (Pittsburgh, PA). Trifluoroacetic acid (TFA) was purchased from Halocarbon (River Edge, NJ). Biotech grade piperidine was from Sigma-Aldrich (St Louis, MO). *N*,*N*-Diisopropylethylamine (DIPEA) was from Applied Biosystems (Foster City, CA). All other reagents were purchased from Sigma-Aldrich at the highest available purity.

#### 4-Iodophenylalanine mutant

The linear precursor was synthesized using Fmoc-chemistry on Wang resin at a 0.25-mmol scale on a CEM liberty microwave-assisted peptide synthesizer (CEM, Matthews, NC) using standard

instrument protocols except a milder coupling cycle was used for the coupling steps involving 4-iodophenylalanine. Following synthesis, the peptide was cleaved from the resin with reagent K for 4 hr at room temperature. The free peptide was then filtered to remove spent resin beads, precipitated with ice-cold diethyl ether, washed twice with ice-cold ether, and the remaining ether was removed under vacuum. The crude peptide was subsequently dissolved in water to a concentration of 0.5 mg/ml, and the pH of the solution was adjusted to 8.1 with N-methylmorpholine. Oxidative folding was performed by the addition of 0.5 mM GSH, and the peptide was allowed to oxidize in the presence of air. The oxidative folding was monitored using analytical scale RP-HPLC and ESI-MS. The cyclized peptide was purified by reverse-phase HPLC with a Vydac C-18 reverse-phase preparative column (Vydac 218TP1022) using a linear gradient of 14% solvent B to 40% solvent B over a 60-min period (solvent A—0.1% TFA in $H_2O$; solvent B—0.1% TFA in 90% MeCN/$H_2O$). The cyclized peak eluted as the major product and was further characterized by analytical reverse-phase HPLC using C18 reverse-phase analytical column (Vydac 218TP5415) and ESI-MS. From 100 mg of crude peptide, the typical yield of purified toxin was approximately 0.6–0.7 mg.

## Diselenide mutant

The linear precursor of the diselenide mutant was synthesized on Wang resin at a 0.25-mmol scale on a CEM liberty microwave-assisted peptide synthesizer (CEM) except the two steps where SeCys was coupled onto the solid phase were performed manually to conserve reagent. The coupling cycles used on the CEM synthesizer all through the synthesis were milder than standard synthesizer cycles. Following a published protocol (*Walewska et al., 2009*), the peptide was cleaved from the resin using reagent K and DTNP, followed by thiolation in presence of DTT. The crude peptide was purified using reverse-phase HPLC with a Vydac C-18 reverse phase preparative column (Vydac 218TP1022) using a linear gradient of 14% solvent B to 40% solvent B over a 60-min period (solvent A—0.1% TFA in $H_2O$; solvent B—0.1% TFA in 90% MeCN/$H_2O$). It was subjected to cyclization as described above for the 4-iodophenylalanine mutant, and the cyclized product was again purified by reverse-phase HPLC with a Vydac C-18 reverse-phase preparative column (Vydac 218TP1022) using a linear gradient of 14% solvent B to 40% solvent B over a 60-min period (solvent A—0.1% TFA in $H_2O$; solvent B—0.1% TFA in 90% MeCN/$H_2O$). The product was further characterized by analytical reverse-phase HPLC using C18 reverse-phase analytical column (Vydac 218TP5415) and ESI-MS. From ~120 mg of crude peptide, the typical yield of purified toxin was approximately 0.6–0.8 mg.

## Crystallization of the toxin–paddle chimera complexes

Concentrated toxin (typically approximately 2–3 mM in water) was supplemented with detergents and lipids so that the final concentration of detergents and lipids in the toxin were the same as in the protein. The concentrated paddle chimera-β2 complex at 8–9 mg/ml was mixed with toxin such that there was a threefold to fourfold molar excess of toxin over the tetrameric channel. Crystallization trials were set up by mixing the toxin-channel complex with the crystallization solution in a 2:1 ratio and supplemented with 10% vol/vol of 40 mM CHAPS. The complexes were crystallized using the hanging drop vapor diffusion method over reservoirs containing 0.1 ml crystallization solution at 20°C. The crystallization solution contained 28–32% PEG400 and 50 mM Tris–HCl, pH 8.8–9.1. Crystals appeared typically within 1–3 days and were directly frozen in liquid nitrogen after overnight equilibration against a reservoir solution containing 33% PEG400 and 50 mM Tris–HCl, pH 8.5.

## Structure determination and building a model for CTX

All diffraction data including selenium and iodine anomalous diffraction data were collected at beamline X29 (Brookhaven NSLS), and images were processed with HKL2000 (*Otwinowski and Minor, 1997*). Data were further processed using the CCP4 suite (*Dodson et al., 1997*). The crystals were isomorphous to the paddle chimera channel, belonging to the P4212 space group. The paddle chimera structural model (PDB ID 2 R9R; *Long et al., 2007*) without ions was used as a starting model. Rigid body refinement in CNS (*Brunger et al., 1998*; *Brunger, 2007*; keeping the same test set as wild type) generated an acceptable model with R-free below 35%. Further rounds of iterative refinement in CNS and manual rebuilding using COOT (*Emsley and Cowtan, 2004*) generated a more complete model with residual R-free below ~28%. Crystals with the selenomethionine CTX derivative diffracted

to the highest resolution of all three heavy atom derivatives and was chosen for subsequent refinement and for building a model for the CTX.

At this stage, a weighted toxin-omit $2F_o - F_c$ electron density map clearly revealed the presence of the toxin at the mouth of the pore. However, the observed electron density map is a superposition of the electron densities for four individual orientations in which the toxin can bind to the tetrameric channel. Moreover, each orientation is only fractionally occupied, thus making the electron density for each orientation inherently weak. Certain secondary structural features in the map were discernible (*Figure 4A*) and allowed approximate placement of a CTX molecule using the available NMR structure of CTX (*Bontems et al., 1991*). However, it was obvious from the outset that this initial model would require additional data to achieve a reasonable level of accuracy. We used anomalous electron density maps from three different heavy atom derivatives of CTX to obtain further constraints that helped us in improving our initial placement of the toxin (see section "Building a model for the toxin" in main text for details). This improved placement was used as a starting point for crystallographic refinement, which was subsequently refined by rigid-body, coordinate-based B-factor refinement in CNS with manual adjustments, where appropriate. During refinement, the interactions between symmetry-related toxin molecules were prevented by using 'igroup' statements in CNS refinement scripts.

In particular, the asymmetric unit contains two α- and β-heterodimers (molecules A and B, see *Figure 3*). The crystallographic symmetry operation thus creates two independent tetrameric channels, each containing four α-monomers and four β-monomers. The model for the toxin was built for the toxin bound to channel A since the electron density for this toxin was much better defined. The asymmetric unit contained one orientation of the toxin modeled at 0.25 occupancy. Consequently, the fourfold symmetry operation generated the other three possible orientations for the toxin molecule to bind to the channel. The independence of molecules A and B in the asymmetric unit and anomalous electron density from the selenomethionine derivative of CTX provided additional validation for our model of the toxin (see section "Building a model for the toxin" in main text for details). Crystallographic data and refinement statistics are shown in data collection and refinement statistics table. Figures were made using PYMOL (www.pymol.org) and COOT (*Emsley and Cowtan, 2004*).

## Structure determination and model building for the Lys27Met CTX–paddle chimera complex and CTX–paddle chimera complex in CsCl

Diffraction data were collected at beamline X29 (Brookhaven NSLS), the data were processed, and the structure was solved as above. An initial omit map calculated using a model without any ions or toxin clearly revealed the presence of the toxin at the mouth of the pore in each case. However, in order to build a model for the toxin in each case, we started from the model of the toxin as built above and carried out subsequent rigid-body, coordinate-based B factor refinements in CNS (*Brunger et al., 1998*; *Brunger, 2007*) with slight manual adjustments of the model in COOT (*Emsley and Cowtan, 2004*) wherever deemed appropriate from prominent features in the electron density map.

## Electrophysiological studies of toxin-channel interactions

Electrophysiology of paddle chimera channels was performed essentially as described (*Tao and MacKinnon, 2008*), except in the experiments for determining $K_d$'s of paddle chimera with wild-type CTX, Lys27Met CTX, and wild-type CTX in the presence of CsCl, the planar bilayer membranes were painted with 20 mg/ml 1,2-diphytanoyl-sn-glycero-3-phosphocoline (DPhPC; *Figure 6A*). For experiments in CsCl, the KCl in the *cis-* and the *trans-*chambers was replaced by CsCl.

The membranes were held at −110 mV and repeatedly pulsed to +110 mV test voltage. CTX inhibition data measured with different concentrations of CTX or Lys27Met mutant of CTX were fit to the equation:

$$I/\text{Imax} = 0.1 + 0.9 \times \left(1 + [\text{CTX}]/K_d\right)^{-1}.$$

The modified form of the equation was used because even at the highest concentration of the toxin, we observed residual current representing ~10% of the initial current that is due largely to the channels facing in the opposite direction. Statistical fit and determination of $K_d$ values were done using the IgorPro (Wavemetrics, Portland, USA) software.

For experiments showing paddle chimera block by a single concentration of heavy atom derivatives of CTX (*Figure 4—figure supplement 1B–D*), the planar lipid bilayers were formed of 20 mg/ml 3:1

(wt:wt) 1-palmitoyl-2-oleoyl-*sn*-glycero-3-phosphoethanolamine (POPE): 1-palmitoyl-2-oleoyl-*sn*-glycero-3-phospho-(1'-*rac*-glycerol; POPG). For the experiment showing paddle chimera block by single concentration of wild-type CTX in *Figure 4—figure supplement 1A*, the planar lipid bilayer was formed of 20 mg/ml 1,2-diphytanoyl-sn-glycero-3-phosphocoline (DPhPC).

## Acknowledgements

We thank Tom Muir (Princeton University) for access to CEM peptide synthesizer and Miquel Vila-Perello in the Muir lab for help and advice on peptide synthesis. We thank H Robinson at beamline X29 (National Synchrotron Light Source, Brookhaven National Laboratory) for assistance at the synchrotron; J Butterwick for comments on the manuscript. RM is an Investigator in the Howard Hughes Medical Institute. Supported by NIH GM43949 awarded to RM. The atomic coordinates and structure factors have been deposited to the Protein Data Bank with the accession numbers 4JTA (selenomethionine CTX with paddle chimera), 4JTD (Lys27Met CTX with paddle chimera), and 4JTC (CTX with paddle chimera in Cesium Chloride).

## Additional information

### Funding

| Funder | Grant reference number | Author |
| --- | --- | --- |
| Howard Hughes Medical Institute | | Roderick MacKinnon |
| National Institutes of Health | GM 43949 | Roderick MacKinnon |

The funders had no role in study design, data collection and interpretation, or the decision to submit the work for publication.

### Author contributions

AB, Conception and design, Acquisition of data, Analysis and interpretation of data, Drafting or revising the article; AL, EC, Acquisition of data, Analysis and interpretation of data, Drafting or revising the article; RM, Head of Lab in which this research was conducted. Conception and design, Acquisition of data, Analysis and interpretation of data, Drafting or revising the article

## Additional files

### Major datasets

The following datasets were generated:

| Author(s) | Year | Dataset title | Dataset ID and/or URL | Database, license, and accessibility information |
| --- | --- | --- | --- | --- |
| MacKinnon R, Banerjee A, Lee A, Campbell E | 2013 | Crystal structure of Kv1.2-2.1 paddle chimera channel in complex with Charybdotoxin | 4JTA; http://www.rcsb.org/pdb/search/structidSearch.do?structureId=4JTA | Publicly available at the RCSB Protein Data Bank (http://www.rcsb.org/pdb/). |
| Banerjee A, Lee A, Campbell E, MacKinnon R | 2013 | Crystal structure of Kv1.2-2.1 paddle chimera channel in complex with Lys27Met mutant of Charybdotoxin | 4JTD; http://www.rcsb.org/pdb/search/structidSearch.do?structureId=4JTD | Publicly available at the RCSB Protein Data Bank (http://www.rcsb.org/pdb/). |
| Banerjee A, Lee A, Campbell E, MacKinnon R | 2013 | Crystal structure of Kv1.2-2.1 paddle chimera channel in complex with Charybdotoxin in Cs+ | 4JTC; http://www.rcsb.org/pdb/search/structidSearch.do?structureId=4JTC | Publicly available at the RCSB Protein Data Bank (http://www.rcsb.org/pdb/). |

The following previously published datasets were used:

| Author(s) | Year | Dataset title | Dataset ID and/or URL | Database, license, and accessibility information |
|---|---|---|---|---|
| Long SB, Tao X, Campbell EB, MacKinnon R | 2007 | Shaker family voltage dependent potassium channel (kv1.2-kv2.1 paddle chimera channel) in association with beta subunit | 2R9R; http://www.rcsb.org/pdb/explore/explore.do?structureId=2R9R | Publicly available at the RCSB Protein Data Bank (http://www.rcsb.org/pdb/). |
| Bontems F, Gilquin B, Roumestand C, Menez A, Toma F | 1992 | Analysis of side-chain organization on a refined model of charybdotoxin: structural and functional implications | 2CRD; http://www.rcsb.org/pdb/explore/explore.do?structureId=2CRD | Publicly available at the RCSB Protein Data Bank (http://www.rcsb.org/pdb/). |

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
