## [Decision Letter]

Thank you for choosing to send your work entitled “Structure of a Pore-Blocking Toxin in Complex with a Eukaryotic Voltage-Dependent K^+^ Channel” for consideration at *eLife*. Your article has been favorably evaluated by a Senior editor, John Kuriyan, and a reviewer, Chris Miller.

The two reviewers discussed their comments before we reached this decision, and the Senior editor has assembled the following comments to help you prepare a revised submission.

**Summary:**

This superb manuscript from MacKinnon and colleagues presents an X-ray structure long sought, but persistently absent, of a scorpion toxin (Charybdotoxin, CTX) bound to a mammalian voltage-gated potassium channel. The channel-toxin system offers a deeply studied example of molecular recognition and thermodynamics and kinetics of protein–protein interactions, as assessed by mutagenesis methods in the absence of direct structural information. And now there is a structure, which dramatically confirms most of the former indirect conclusions, and provides deeper and more elaborate interpretations to them.

The toxin binds to the channel in a 1:4 stoichiometry between toxin and channel subunits. In order to determine the structure, the authors had to overcome a formidable challenge that arises due to the mismatch between the four-fold symmetry in the channel and the lack of symmetry in the toxin. They did this by incorporating three different heavy atom probes in the toxin, and measuring X-ray anomalous diffraction for crystals with each derivative. In this way, they were able to place the toxin at the mouth of the pore with a reasonable degree of confidence. The structure has been determined at 2.5 Å resolution, which leads to further confidence in the interpretations presented here.

While the key feature of the toxin blocking mechanism–the key Lys27 sidechain acting as the closest thing that nature could provide to a tethered K_ ion–is too low in density to be directly observed here, its presence is loudly announced by the disposition of K^+^ ions in the selectivity filter. The authors have cleverly argued from these ion occupancies–and extended these arguments by using the variation in occupancy seen with Cs^+^ – for the expected disposition of Lys27.

This paper is important because, as the authors point out, the analysis of toxin–channel interactions has provided key insights into the structure and mechanism of K^+^ and other channels since the very beginning, but no crystal structure is available (until now) for the complex. Thus, even though most of what is described here is consistent with what was inferred previously, the availability of this new structure provides a reliable framework for the design of new experiments and the interpretation of new ones. Furthermore, NMR data (cited in the paper) had been used previously to infer that the channel undergoes conformational changes upon toxin binding. The crystal structure presented here demonstrates that it does not, and shows that the binding is well described by a classic “lock and key” interaction. This clears up a possible source of confusion and adds to the significance of this paper.

The paper is very well written and clearly illustrated. Publication in *eLife* is recommended essentially as is, but the following minor points could be considered while finalizing the manuscript for publication:

1) This well-argued paper could perhaps be improved somewhat if the very long run-on paragraphs are broken up to allow the reader to follow the discussion better.

2) The figures are notable for their clarity, but the addition of a few more labels could provide guides for the non-specialist reader to follow more readily. For example, in Figure 1, identify the structural module colored blue. In Figure 2, point out where the channel subunit is, and where the beta subunit is. We don't mean to overemphasize this point, but the authors may consider readers who are not overly familiar with the architecture of the voltage-gated channels.

3) The Introduction should point out that the structures of the scorpion toxins (mention the important ones) are very similar, and they are expected (known?) to bind the channels in similar ways. This point is key to the arguments in the Discussion, but is taken for granted. In particular, the concluding part of the paper says that: “The CTX family of K^+^ channel toxins has a remarkable diversity in sequence and binding affinities to specific channel subtypes.” Given this, might the reader not assume that the binding modes might be different, and that some toxins might require conformational changes and others might not?

4) A crystallographer might find it helpful to see a top-down view along the four-fold axis of the actual (pre-modeling) electron density map, and the refined map after modeling. This would help understand the statement that initially “certain secondary structural features were discernible…”

5) When discussing the Kaliotoxin NMR “structure”, the authors might state whether or not the contacts inferred from the NMR data are consistent with the crystal structure.

6) Could the authors comment on why active CTX requires its N-terminus to be blocked with pyroglutamate, if any hints of this emerge from the structure?

---

## [Author Response]

*1) This well-argued paper could perhaps be improved somewhat if the very long run-on paragraphs are broken up to allow the reader to follow the discussion better*.

We have revised the manuscript, especially the Introduction and Discussion, to improve readability. Numerous paragraphs have been shortened and tightened.

*2) The figures are notable for their clarity, but the addition of a few more labels could provide guides for the non-specialist reader to follow more readily. For example, in Figure 1, identify the structural module colored blue. In Figure 2, point out where the channel subunit is, and where the beta subunit is. We don't mean to overemphasize this point, but the authors may consider readers who are not overly familiar with the architecture of the voltage-gated channels*.

Labels have been added as requested to Figure 1, as well as a shaded area to demarcate the membrane, two additional toxins added to Figure 1 (as requested below), coloration to distinguish components in Figure 3 (we believe the reviewer was referring to Figure 3 in the comment about Figure 2 above), we have changed the orientation in Figure 4 to show secondary structure elements in toxin density (as requested below), and we have added ion site labels S1–S4 on Figure 6.

*3) The Introduction should point out that the structures of the scorpion toxins (mention the important ones) are very similar, and they are expected (known?) to bind the channels in similar ways. This point is key to the arguments in the Discussion, but is taken for granted. In particular, the concluding part of the paper says that: “The CTX family of K^+^ channel toxins has a remarkable diversity in sequence and binding affinities to specific channel subtypes.” Given this, might the reader not assume that the binding modes might be different, and that some toxins might require conformational changes and others might not*?

We have addressed this point by adding to the Introduction the sentences below (in quotes) and we now show three different toxin structures in Figure 1:

“The positive charge was later identified as Lys27, a residue that is conserved in all members of the CTX‐like toxin family (Figure 2) (48; 19). Studies with other members of the CTX toxin family, most extensively Agitoxin2 (AgTx2), supported the conclusion that they function in a manner similar to CTX (16; 29; 24; 49). Most notably, conservation of toxin shape and the functionally important lysine suggested they all bind with a similar orientation on a K^+^ channel and inhibit through a common mechanism whereby a lysine amino group functions as a K^+^ ion mimic to block the pore (38) (Figure 1).”

*4) A crystallographer might find it helpful to see a top-down view along the four-fold axis of the actual (pre-modeling) electron density map, and the refined map after modeling. This would help understand the statement that initially “certain secondary structural features were discernible…*”

We have addressed this point by showing a different orientation of toxin omit map density in Figure 4. This makes the point better than the suggested view down the four-fold axis.

*5) When discussing the Kaliotoxin NMR “structure”, the authors might state whether or not the contacts inferred from the NMR data are consistent with the crystal structure*.

This is a very important point that we did not address explicitly, but in response to this comment we have made modifications (below). Basically, the solid state NMR data did not contain distance restraints between toxin and channel, and therefore the origin of the “structure” is entirely unclear. The paper simply shows some chemical shift changes in both the toxin and channel upon binding, which very possibly do not have their origin in structural changes. Thus, we have removed the reference to Lange et al. from the last paragraph of the Discussion (and kept the references to Takeuchi et al. and Yu et al., which refer to NMR models based on distance restraints). Regarding the conformational changes proposed in the Kaliotoxin NMR “structure” (Lange et al.) we have added the following to the Discussion:

“This, we note, is in contrast to the solid state NMR ‘structure’ of a KcsA mutant with Kaliotoxin, where such rearrangements in the channel were proposed (30). In this NMR study chemical shift changes induced by toxin were interpreted as resulting from dihedral angle changes (i.e., structural); however, such chemical shift changes could have other origins (i.e., electrostatic). Moreover, no channel–toxin distance restraints were included in the determination of this solid state NMR ‘structure’ (30). In the crystal structure presented here…”

*6) Could the authors comment on why active CTX requires its N-terminus to be blocked with pyroglutamate, if any hints of this emerge from the structure*?

In order to say something conclusively about this, we would need more mutagenesis results with same channel-toxin pairs, which we do not have (i.e., mutations on the N-terminus have been done with toxin-channel pairs that are different than our structure and therefore we would have to speculate on the basis of homology modeling). We’d rather not. However, for purposes of intellectual exchange we offer the following thoughts here should the reviewers want to think about it for future work:

The most subtle experiment has been done in the Goldstein, Pheasant, and Miller paper in Neuron (1994) where the pyroGlu was mutated to a proline resulting in a near-isosteric but charge inequivalent substitution. The result was an 8-fold increase in Koff and a 570-fold change in K_d_. Now if one looks at the region of the channel where the pyroGlu approaches the channel and compares an alignment between Kv1.2 and Shaker, here is what we see –

Kv1.2 …D E R D S Q F P…

Shaker …G S E N S F F K…

The underlined residue, which is Q in the structure and is Phe in Shaker. In one conformation of the Phe side chain, it could possibly get into disfavourable interactions with a charged terminus. That would explain the results with Shaker.

(This implies that with the same pyroGlu to proline mutation and with paddle chimera the effect of the pyroGlu to proline mutation would be less–information that we don't know about.)

The other piece of information is with BK from the Park, Hausdorff, and Miller paper in PNAS (1991) where the uncyclized material shows an apparent K_d_ of 110 nM as compared to 10 nM for the native toxin (Anderson, MacKinnon, Smith and Miller, J. Gen. Physiol., 1988). However, here there may be dual effects from charge and sterics as well. Comparison of the same region for Kv1.2 and BK looks like the following –

Kv1.2 …D E R D S Q F P…

BK …S G D P L D F D…

However, these two regions are much harder to align between Shaker and BK and it is harder to rationalize smaller changes in affinity based on this. Thus, we have chosen to not to try to interpret these results.